

# Ionospheric Plasma Density Measurements by Swarm Langmuir Probes: Limitations and possible Corrections

Piero Diego[1], Igino Coco[2], Igor Bertello[1], Maurizio Candidi[1], Pietro Ubertini[1]

[1]Istituto Nazionale di Astrofisica, via del Fosso del Cavaliere 100, Rome, Italy.

[2]Istituto di Geofisica e Vulcanologia, via Vigna Murata, Rome 605, Italy.

Correspondence to: Piero Diego (piero.diego@inaf.it)

**Abstract.** The ESA Swarm constellation includes three satellites, which have been observing the Earth's ionosphere since
November 2013, following polar orbits. The main ionospheric plasma parameters, such as electron density and temperature, are measured by means of Langmuir probes (Lps); electron density measurements, in particular, are nowadays largely considered as qualitatively reliable, and have been used in several published papers to date. In this work, we aim to discuss how some technical characteristics of Swarm Lps, such as their size and location on board the satellites, as well as the operational setup of the instruments, could lead to limitations in their accuracy if one underestimates the influence of
satellite proximity, and the larger extension of the plasma sheath surrounding the probes due to the operational point of the voltage ripple. Two specific corrections are proposed for the assessment and possible mitigation of such effects. Finally, a comparison is made with electron density measurements from CSES-01 mission, which relies on Langmuir probes as well, whose geometry and operating mode are standard.

## 1 Introduction

Plasma density and temperature represent the two principal parameters that characterize the state of the ionosphere. Indeed, many observations, either direct and *in-situ* or indirect and remote, have sought after such parameters [Hargreaves, 1992; Schunk and Nagy, 2009; Lomidze et al., 2017]. In addition to groundbased instruments such as incoherent scatter radars (ISRs), ionosondes, and Global Positioning System (GPS) receivers, Lps on board satellites represents a standard for the measurement of the ionospheric electron plasma density *in situ*.

One of the most recent missions mounting Lps is Swarm (Friis-Christensen et al., 2006), launched in November 2013 and comprising three spacecraft, two of which flying in pair and following a free polar orbit at about 470 km (Swarm A and C), and one following a higher free polar orbit at about 520 km (Swarm B). The main objective of Swarm is the mapping of the Earth's magnetic field with unprecedented accuracy but, due to its peculiar orbit configuration and versatile and accurate instrumentation on board, the mission has become a key reference for the magneto/ionosphere science community as well.

The Electric Field Instrument package, including Lps, is described in Knudsen et al. (2017).

So far a number of studies have addressed the comparison between Lp based electron density measurements and data from other instruments/techniques [McNamara et al., 2007; Pedatella et al., 2015a, 2015b; Rother et al., 2010; Lomidze et al., 2017], helping assess the reliability of new data and their consistency with operational ground- and space-based instruments. McNamara et al. (2007) compared CHAMP Lp to Jicamarca ionosonde plasma frequencies, determining that Lp values were





systematically lower by ≈4.2%. In addition, Pedatella et al. (2015a) compared CHAMP Lp observations to COSMIC GPS radio occultation (RO) density measurements, finding that that COSMIC densities were, on average, greater by ≈14.9%, even though with a correlation coefficient larger than 0.9. Finally, Lomidze et al. (2017) compared Swarm Lp data to ISR measurements (several low-latitude ionosonde stations), and to COSMIC RO data covering all latitude ranges, and found a systematic underestimate of Swarm-derived plasma frequency of about 10% despite the associated high correlation

coefficient.

However, the comparison between *in-situ* and remote measurements is always a very delicate matter: different sizes of sensing areas, different time resolutions, and theoretical models, all involved in the inversion processing of remote measurements, make such comparisons qualitatively reliable only over large dataset statistics, while single case comparisons can lead to big differences.

Therefore, the one-to-one comparison between *in-situ* measurements is always preferable, when available. On February 2018, the first China Seismo-Electromagnetic Satellite (CSES-01) was launched, with the aim to detect electromagnetic perturbations in the ionosphere following earthquakes [Shen, X., et al., 2018]. The satellite follows a polar Sun-synchronous orbit with ascending node at 2-AM local time at about 510 km, and during August 2018 good conjunctions were found with the lower Swarm pair (Swarm A and C). CSES-01, as well as Swarm, mounts Langmuir probes for the measurement of

electron density and temperature [Yan, R., 2018, Guan Y-B., 2018].

As detailed in the following sections, the basic theory retrieving plasma parameters from Lp raw current still relies on the seminal work by Mott-Smith and Langmuir [1926]. However, any algorithm must properly take into account all relevant construction features of the apparatus, as well as adjustments to plasma regimes one expects to encounter along the spacecraft path [Brace, 1998].

In this paper, we analyse some features of the Swarm Lp setup and operations, which, considering the properties of ionospheric plasma at its altitude, are likely to introduce a variable bias in the electron density determination. Correspondingly, we suggest some simple correction algorithms aimed at the mitigation of such effects. In order to justify the reliability of the proposed corrections, we show some case comparison involving density measurements taken by CSES-01 in local-time conjunction with Swarm A.

**2 The Langmuir Probe: general theory**

In general, a Langmuir probe is an electrode immersed in a plasma. If a sweeping voltage is applied to the electrode, a plasma current will be collected [Mott-Smith and Langmuir, 1926], according to a typical current-voltage (I-V) characteristic curve shown in Figure 1.



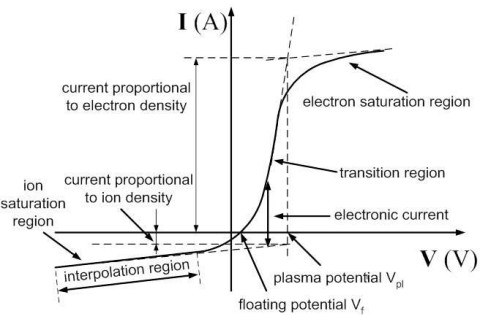

*Figure 1: Current-Voltage characteristic curve of a Langmuir probe (from Diego et al., 2017a)*

For each applied potential, the total current, *I*, is due to the contribution of both ions ("ion current", $I_i$) and electrons ("electron current", $I_e$). As far as the photoelectron emission is negligible (i.e., up to an altitude of several hundreds km in the ionosphere, as described in Diego et al. (2017)), the net current drawn from the plasma is the sum of $I_e$ and $I_i$. The amount of current drawn for each plasma component depends on plasma parameters (mainly density and temperature) and on the ambient magnetic

field to a lesser extent. The interaction between probe and plasma is developed through the so-called "plasma sheath" region, which is formed close to the probe's conducting surface, and whose extension depends on the sweeping voltage. The Lp standard mode of operation consists of sweeping the probe potential widely across the whole characteristic curve (Figure 1), with a voltage modulation, *V* - $V_{pl}$ ($V_{pl}$ being the local plasma potential, and *V* the probe potential), that swings from very negative to very positive values.

The ion and electron densities and temperatures can be obtained by analysing the *I-V* curve of the probe, using the Orbital Motion Limited (OML) theory [Mott-Smith and Langmuir, 1926]. According to OML, for a spherical Lp, the collected currents depend on the sign of the applied voltage with respect to plasma potential: when $q(V - V_{pl}) < 0$, *V* is usually referred as the "retarding" potential, while, in the opposite case, one refers to *V* as the "accelerating" potential, with $q = +/- e$ depending on particle species (ions/electrons).

The electron current collected by an electrode submerged in plasma under retarding potential is given by:

$$I_e = \frac{1}{4} qn \sqrt{\frac{8kT_e}{\pi m_e}} S_e e^{\frac{q(V-V_{pl})}{kT_e}} \quad (1)$$

where *q* is the elementary charge, $S_e$ the cross-section area for electron collection on the probe, and $n = N_e$ the electron density.

Thermal velocity is assumed isotropically distributed in space, even in the presence of a magnetic field if the relevant gyroradius is larger than (or of the same order of magnitude as) the probe radius ($R_p$); taking the average variation range of *B* and $T_e$ for most ionospheric conditions, electron Larmor radius turns out to be several *cm*, which is larger than, or at least comparable to, the radii of most commonly deployed Lps. As a consequence, the probe cross-section, $S_e$, coincides with the surface of a sphere with radius $R_p$, $S_e = 4\pi R_p^2$.

For an accelerating potential (*V* - $V_{pl}$ > 0), two possible conditions, which depend on plasma sheath's relative size with respect to $R_p$, can occur: for plasma sheaths thinner than $R_p$ ("thin sheath" approximation), the collected current tends to get flattened to a constant value corresponding to the thermal current; for sheaths thicker than $R_p$ ("thick sheath" approximation), $I_e$ tends to increase linearly with *V* [Schott, 1968].



The Debye length, $\lambda_D$, within the ionospheric range of electron temperatures and densities at Swarm altitude, is computed to vary between 0.2 cm and 4 cm. Considering that the actual sheath dimension is equal to several $\lambda_D$, minimum sheath thickness can be assumed of the order of one centimetre. Even considering minimum sheath thickness and the standard size range of most common Lps (from few mm to few cm), thin sheath condition is never satisfied, which means that thick sheath regime can be safely applied to the entire range of ionospheric conditions.

With these assumptions, $I_e$, is rewritten as:

$$I_e = \frac{1}{4} qn \sqrt{\frac{8kT_e}{\pi m_e}} S_e \left(1 + \frac{q(V - V_{pl})}{kT_e}\right) \qquad (2)$$

Unlike electrons, in the satellite's reference system, ions are seen as a flux of particles coming from the ram direction with a velocity nearly equal to that of the satellite ($v_i = 7.6 \cdot 10^3$ m/s), since, due to their much larger masses, their thermal velocities are almost negligible in comparison to ram velocity.

Therefore, the space distribution of ion velocity implies that probe cross section for ion collection is that of a flux tube almost aligned with the satellite velocity vector, whose dimensions also depend on probe voltage, being $S_i = \pi R_p^2$ when the probe is at local plasma potential. Even taking into account ion drift, the maximum divergence from the ram direction is lower than 15°, as shown in Diego et al (2017b), where Swarm Thermal Ion Imager (TII) measurements have been used to identify the actual ion arrival direction. Nevertheless, for our analysis, the current collected by the probe can be estimated assuming a flux of mono-energetic ions with energy equal to $K_{ion} = 1/2 \cdot m_i \cdot v_{orb}^2$.

Relying on the same considerations as those for electron collection, we assume here a thick sheath approximation for all ionospheric conditions encountered along the orbit; thus, the ion current can be computed from angular-momentum conservation of particles. The collected current (due to a mono-energetic beam of ions) can be expressed as:

$$I_i = \pi R_p^2 q n v_{orb} \cdot \left[1 - \frac{2q(V - V_{pl})}{m_{ion} v_{orb}^2}\right] \qquad (3)$$

This equation is valid for both retarding and accelerating potentials.

### 2.1 The Swarm Langmuir probe: Instruments and measurement modes

The Swarm Langmuir probes are two equal spheres of 4-mm radius mounted on ~10-cm-long stubs, which are Earth-facing at the front end of the satellite: one, the so called "high-gain" probe, is covered by a thin film of TiN; the other (the "low-gain" probe) is covered by a thin film of gold. Currently, only the high-gain probe is used to determine electron density, and, in the majority of cases, electron temperature as well. The low-gain probe is involved in the determination of the spacecraft potential, $V_s$. Figure 2 is taken from Knudsen et al. (2017), and schematically sketches the principle of the "harmonic" mode, i.e., Swarm Lp's operational mode for 99% of measuring time.



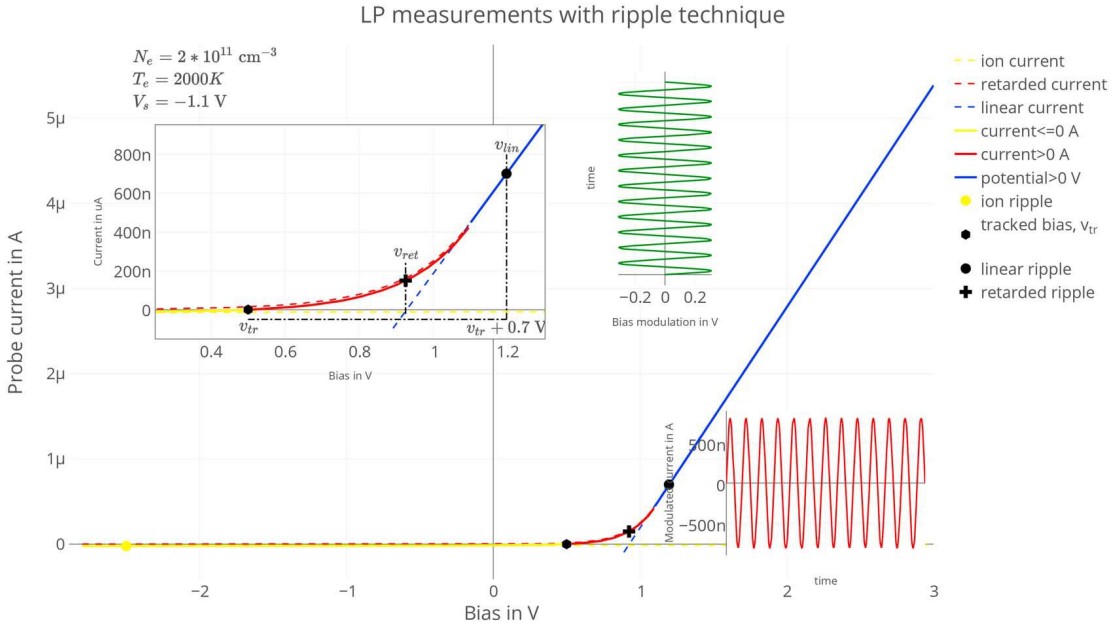

*Figure 2: Swarm Lp ripple technique (from Knudsen et al., 2017).*

Instead of performing a full *V* sweep for exploiting the nominal *I-V* characteristic curve, three bias points are chosen: one in the "ion" region, i.e., at a potential negative enough to ensure that the collected current is almost entirely due to ions ($V_{ion}$); one in the linear region, i.e., at a potential positive enough ($V_{lin}$) to ensure that the collected current is almost entirely due to electrons ("accelerating potential" region); one in the "retarding potential" region, i.e., around the knee of the I-V curve ($V_{ret}$). Bias potentials (always set to -2.5 V in the harmonic mode) are modulated by sinusoidal ripples at frequencies up to about 4

kHz, with a common nominal value of 128 Hz, and with adjustable amplitude (0.2 V in the ion region). Through this harmonic modulation, both the current and complex admittance are obtained. The real part of the admittance is the derivative of the *I-V* characteristic curve. At each point, the "rippling" has a duration of the order of 100 ms, and the current and admittance are averaged over roughly ten ripple cycles. In this way, three pairs of *I* and *dI/dV* values can be recorded each half second, from which $N_e$, $T_e$ and $V_s$ can be derived.

In Knudsen et al. (2017) is shown how, starting from the Mott-Smith and Langmuir theory, and under the hypothesis of plasma neutrality ($N_e = N_i$), electron density can be retrieved using the admittance of the ion region only, $D_{ion}$, as follows:

$$D_{ion} = \pi R_p^2 q v_i \, n / E_i, \qquad (4)$$

where Rp is the probe radius, q the elementary charge, $v_i$ the ram ion speed (which, as explained in Sect. 2, can be considered

equal to $v_{orb}$, i.e, the spacecraft velocity), and $E_i$ the kinetic energy of the ion flow, that is, $E_i = \frac{1}{2} m_i v_{orb}^2$. A further assumption is that the only ion species present at any Swarm altitude is O+, so that $N_i = N_{O+}$ and $m_i = m_{O+}$. This assumption is reasonable under most conditions, but a certain amount of H+ contamination cannot be occasionally excluded, especially at high latitudes and during geomagnetically active periods.





Electron density obtained in such a way is considered very stable, and qualitatively reliable, since it reproduces all the expected macroscopic features of the ionospheric density known from the past literature. In addition, since its launch, several studies have used Swarm Lp data to analyze various physical phenomena connected to ionospheric dynamics [e.g.: Buchert et al., 2015; Goodwin et al., 2015; Park et al., 2015, 2016].

## 3 Probe polarization-induced effects

As shown in the previous section, Swarm Lps determine $N_e$ by an operational setup including a fixed -2.5 V bias set on the probe, and +/- 0.2 V swings. In this section, we want to highlight two different effects induced on plasma surrounding a probe when a fixed negative polarization is applied:

1) Such negative bias can cause a plasma sheath enlargement that actually modifies the particles collection by the sensors.

2) If the probe is too close to satellite surface, and operates in conditions under which plasma neutrality cannot be achieved, an electric field may appear in the proximity of the satellite surface, which, in turn, increases the effective cross section of the probe.

compensation algorithms to evaluate and mitigate both perturbations in the measurement of $N_e$ are presented in the next subsections.

### 3.1 Swarm Plasma-sheath effect

As known [Bhattarai and Mishra, 2017, and references therein], a conductive body immersed in a plasma, when polarized at negative potential, is surrounded by a plasma sheath filled with ions. The sheath size is about $5\lambda_D$ [Chen, 1984], therefore, at high latitudes where $\lambda_D$ increases, it can reach a diameter much larger than $R_p$. As a matter of fact, a sensor surrounded by a large sheath collects an additional quantity of ions, widening the collection region for ions beyond the mere cross sectional area of the probe, even though the actual plasma density is decreased producing a saturation-like effect.

The correction factor is obtained considering the effective collecting surface equal to the sheath size [Chen, 2001], such that, starting from Eq. 3, one can define an "effective" current collected by the probe as:

$$I_{eff} = \pi R_p^2 q n V_{orb}\left(1 + \frac{\sqrt{2}}{3 r_p}\ S_h\ \frac{q|V - V_{pl}|}{K_{ion}}\right)\ (5)$$

where $S_h \approx (5*\lambda_D)/R_p$. In this way, the correction factor, $CF$, is rewritten as

$$CF = I_{eff}\ /\ I_{ion} = \frac{\pi R_p^2 q n v\left(1 + \frac{\sqrt{2}S_h}{3R_p}\frac{q|V - V_{pl}|}{K_{ion}}\right)}{\pi R_p^2 q n v\left(1 + \frac{q|V - V_{pl}|}{K_{ion}}\right)},\ (6)$$

and can be easily evaluated assuming:

$V = -2.5$ V

$V_{pl} = hkT_e/q$     with respect to floating potential

$K_{ion} = \frac{1}{2} \cdot m_i \cdot v_{orb}^2$

$S_h \approx (5 \cdot \lambda_D)/R_p \propto \text{sqrt }(T_e/N_e)$



where $v_{orb} \approx 7600$ m/s, $k$ the Boltzmann constant, and $h$ a constant depending on the ion species that surround the probe
[Merlino, 2017]. In our case, at Swarm orbit, $h$ ranges from 3 (only Hydrogen ions) to 5 (only Oxygen ions). Here, we
consider an average of 4.

Since a linear correlation exists between collected current and electron density, applying a correction factor to the current is
equivalent to applying it to the density. Therefore:

$N_e^* = N_e /CF$          (7) ,

where $N_e^*$ can be considered as the "true" electron density inferred from a measured current located well into the ion region,
where the plasma sheath surrounding the probe results enlarged up to $5*\lambda_D$, as described above. One should note, however,
that both $N_e$ and electron temperature, $T_e$, are included in the calculation of the correction factor, as both quantities appear
in the determination of the Debye length: while, in principle, one should use the "true" density (e.g., a density retrieved from

independent measurements performed by other missions/instruments nearby, in which density is not evaluated in the ion
region), at a first approximation only a number of non-corrected density measurements are available. The correction factor
is, therefore, to be interpreted in a qualitative sense. Moreover, considering that the enlargement of the Debye sheath due to
the negative bias applied to the probe could lead to an overestimate of the electron density, at worst an underestimate of the
$CF$ is expected.

**3.2 Effects from Swarm Satellite potential**

A spacecraft (S/C) is a floating body immersed in a plasma, which means that it collects ions and electrons, and emits
photoelectrons depending on several parameters such as S/C shape, orbital velocity ($v_{orb}$), electron temperature, etc.
Starting from the (generally unknown) plasma potential, which we can set to null, the floating body's potential varies in
order to adjust currents until their sum is equal to zero.

Due to thermal velocity of electrons, this floating potential is usually negative compared to plasma potential at the altitude of
Swarm where the photoelectron contribution is negligible.

In Figure 3, the $V_{s/c}$ provided by Swarm Lps is shown for a sample dayside (panel a) and nightside semi-orbit (panel b) on
August 20, 2018 (Solar Quiet, SQ, day); red dashed lines represent the Lp operating bias voltage of -2.5 V in the ion collection
region. Along daylight paths, $V_{S/C}$ clearly increases at low latitudes according to plasma-density enhancement, thus becoming

more positive compared to probe bias by up to about 1 Volt (about -1.5 V compared to the -2.5 V). Conversely, along
nightside paths, $V_{S/C}$ enhancement is observed at mid latitudes, though reaching, on average, about the same level as dayside
values. Every time $V_{S/C}$ becomes positive with respect to the probe bias voltage, an electric field is created between the probes
and S/C body.

In case of a short distance between probes and S/C, a rather strong electric field is expected to take place. For a Swarm Lp,

this distance is less than 10 cm, causing the electric field to be of the order of 10 V/m with the 1 V difference mentioned right
above. In addition, this electric field is perpendicular to the ion ram direction, so it proves to be very effective in deflecting
charged particles from their straight trajectory. This means that a portion of particles, nominally outside the cross section of
the sensor, is actually captured in the space between S/C and probes. This non-symmetric electric field produces an unbalanced
enhancement of the ion flux towards the probes.





In order to assess whether particle deflection is significant, we have to consider how relevant is the time of flight (ToF) of the
ions in the space between the Lp and S/C, at the ion ram velocity of about 7600 m/s.

Any Swarm Lp usually collects ions on a surface perpendicular to the ram direction, that is, a disk of radius $R_p$.

Under the hypothesis of negligible drift ion velocity, in the satellite's reference system the arrival direction of ions is along
its orbital velocity $- v_{orb}$, thus one can easily evaluate the enlarged probe cross section as a function of the electric field $E$
induced by the difference between Lp and S/C potentials along the orbit, which translates into a "corrected" or effective radius
of the probe:

$$R_c = R_p + \frac{qE}{2m} t^2 \qquad (8),$$

with

    $m$=2.5·10$^{-26}$ kg (Ionized Oxigen mass), and $t \approx$10 μs the traveling time in the space between S/C and Lp.


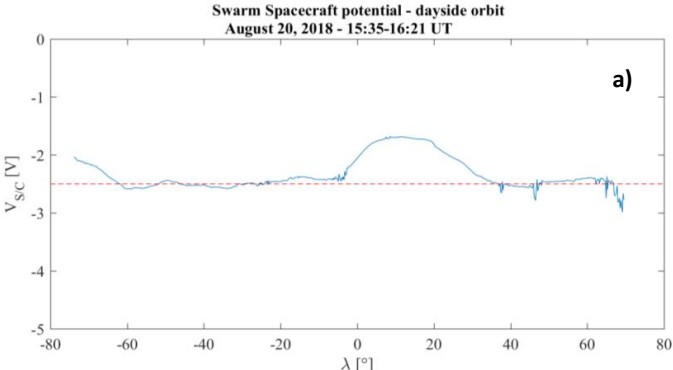

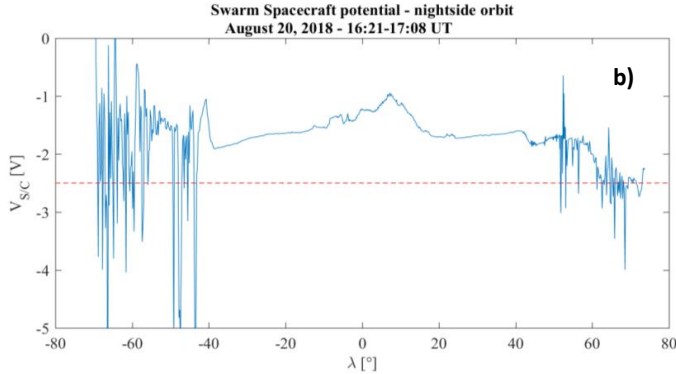

*Figure 3: Swarm spacecraft potential for August 20, 2018 along dayside (panel a) and nightside (panel b) semi-orbits as a function of
geographic latitude. Red dashed lines represent Lp operating bias voltage at -2.5 V.*





**4 Swarm – CSES comparison**

In order to qualitatively test the efficiency of the proposed corrections, we compare density measurements from Swarm Lps to those obtained from the Langmuir Probe experiment (LAP) on board CSES-01. LAP operates in accordance with the classic theory by Mott and Smith, and flew almost at the same altitude and local time of Swarm lower pair (A and C) in August 2018 (see example in Figure 5). LAP consists of three parts: sensors (probes), booms and the electronics box [Liu et al., 2019]. The sensors are spheres with diameters of 1 cm (Lp1) and 5 cm (Lp2), respectively.  In each probe, the upper hemisphere is the

collecting electrode, while the lower one is a guard electrode to which the same voltage as the upper one is applied. Only the currents collected by the upper electrode are used for the computation of plasma parameters, whereas the guard is needed to keep a symmetrical potential distribution around the probes. Probes and guards are made of Titanium, and are coated with Titanium Nitride (TiN). In order to prevent effects from the S/C potential, each spherical probe is mounted on a 50-cm long boom. This causes the probes to be outside the plasma sheath surrounding the spacecraft. In addition, LAP probes are

positioned on the ram side of the spacecraft, with the booms along S/C velocity direction. Therefore, any residual ion flow bending, due to S/C potential, can be neglected. The current collected by Lp2 is much larger than the one detected by Lp1, due to the larger particle collecting area of the former probe suitable for lower current measurements. The sweeping voltage is in the range [-6:6] V in the case of LP1, and [-3:3] V in the case of Lp2. The period of the sweeping voltage is 3 s (including forward and reverse sweeping).


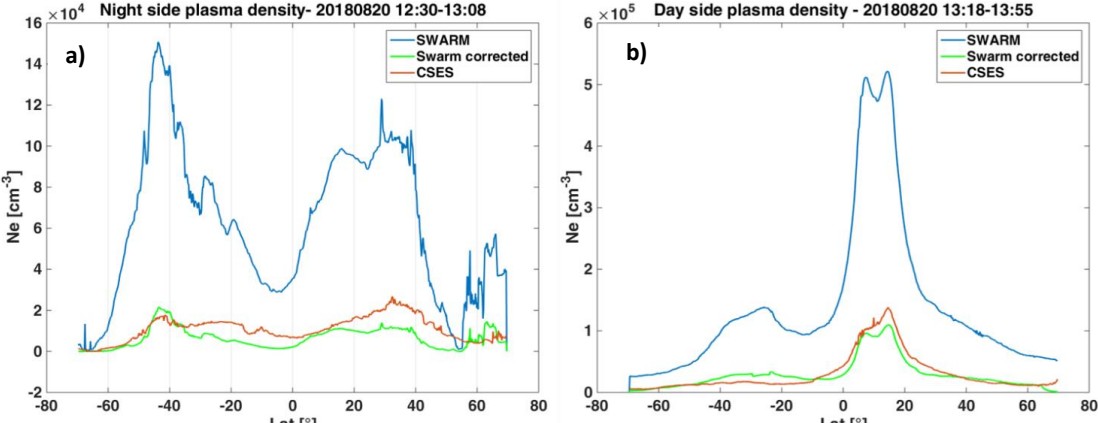

*Figure 4: Superposition of Swarm plasma density (blue), Swarm corrected plasma density (green), and CSES-01 observations (Red) for nightside (panel a) and dayside semi-orbits (panel b) as a function of latitude on Aug 20, 2018 (SQ day).*

Figure 4 shows the superposition of Swarm plasma density (blue), Swarm corrected plasma density (green), and CSES-01

observations (Red) for nightside (panel a) and dayside semi-orbits (panel b) as a function of latitude on Aug 20, 2018 (SQ day). As shown, in spite of the expected differences in electronic density (≈35%) due to different satellite altitudes (about 30 km on average) and different orbit inclinations, $N_{e,S}$ (where S stands for Swarm) is about 4 times  as large as $N_{e,C}$ (where C stands for CSES-01) at low latitudes, and up to 20 times as large at high latitudes. This latitudinal variation depends on the typical $N_e$ reduction at higher latitudes, which implies an increase in Debye length that is generally the key parameter for

plasma instrument setting and measurements [Chen, 1984, Merlino, 2007].



Once the algorithms described above are applied, $N^*_{e,S}$ (green line) gets dramatically reduced, becoming comparable to $N_{e,C}$. The algorithm for sheath effect (Sect. 3.1, Eq. 7) is particularly efficient at higher latitudes (i.e., for lower densities and longer Debye length). On the other hand, the S/C-probe electric-field correction (Sect. 3.2, Eq. 8) becomes important at low and medium latitudes, where S/C voltage turns positive with respect to fixed probe potential of -2.5 V in the ion collection region.

The remaining differences between $N^*_{e,S}$ and $N_{e,C}$ over both dayside and nightside semi-orbits can be related to the different orbit inclination of the two satellites. Figure 5 shows Swarm (blue) and CSES-01 trajectory (red) corresponding to plasma density observations reported in Figure 4. Indeed, the different shapes of the two semi-orbits clearly explains why Swarm observed a quasi-symmetric plasma density double peak, while CSES-01 detection resulted in a greater peak in the northern equatorial region: unlike CSES-01, Swarm crosses the equatorial region almost along the same meridian.


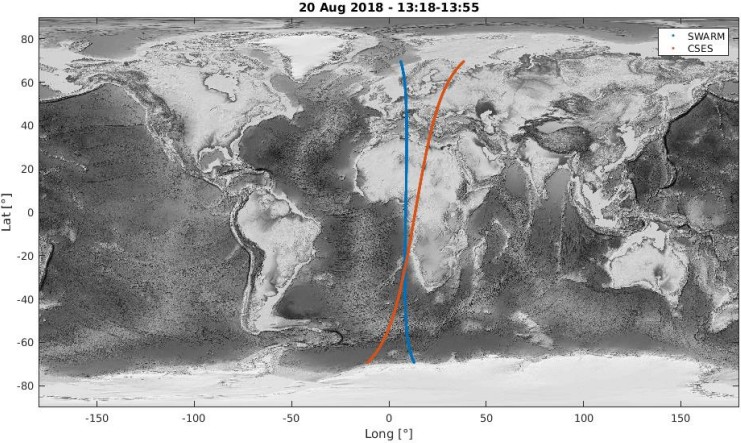

*Figure 5: Swarm (blue line) and CSES01 semi-orbit (red line) as a function of geographic latitude and longitude between 13:18 UT and 13:55 UT on August 20, 2018. The planisphere has been derived from an image of NOAA (doi:10.7289/V5C8276M)*

Figure 6 shows the scatter plots of $N_{e,c}$ vs $N_{e,s}$ before (panel a) and after (panel b, $N^*_{e,s}$) the application of our correction

algorithms to the dayside semi-orbit shown in Figure 4 for the SQ day of Aug 20, 2108. As confirmed by the red fit ($N_{e,C} = m \cdot N_{e,S}$), linear scaling of $N_{e,C}$ with respect to $N_{e,S}$ along the bisector is poor (angular coefficient m = 0.22), while $N_{e,S}^*$ and $N_{e,C}$ are much better related (m = 1.11).



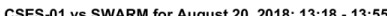

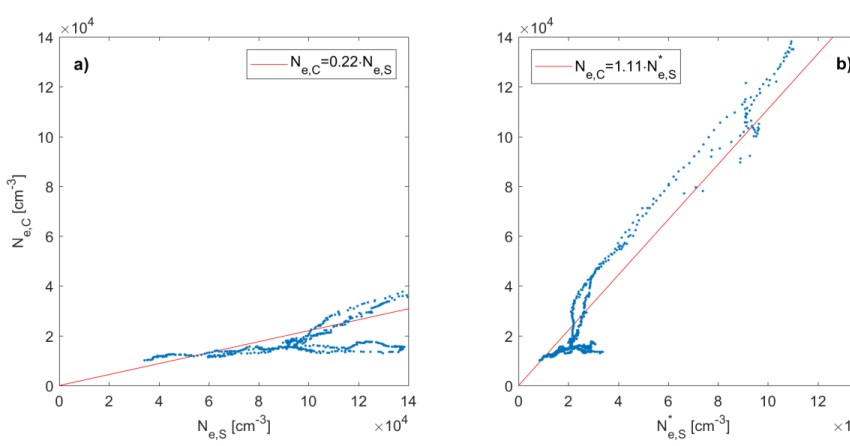

*Figure 6. Scatter plot of CSES-01 electron density as a function of Swarm electron density before (panel a) and after (panel b) the application of correction algorithms for the August 20, 2018 semi-orbit between 13:18 UT and 13:55 UT; red line is the linear fit.*


Figure 7 shows the results of the comparison between $N_{e,S}$ and $N_{e,C}$ for the entire SQ day of August 20, 2018 in terms of $P=N_{e,S}/N_{e,C}$ ratio along dayside (panel a) and nightside semi-orbits (panel b). Blue histograms represent algorithm-uncorrected P values, and confirm that Swarm observed, on average, densities 4 times as large as CSES-01's along dayside

semi-orbits and 5 times as large along nightside ones. The situation dramatically changes when the correction algorithms are applied. Indeed, on either side of the planet, P (red histograms) is about 1 on average, confirming that Swarm and CSES-01 are now observing the same plasma densities.



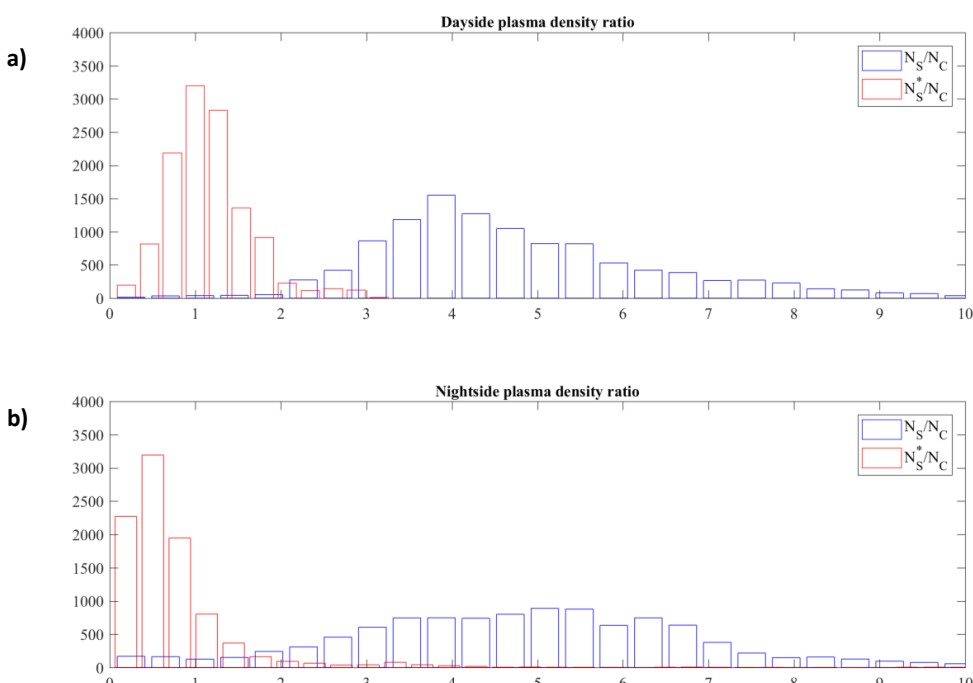

*Figure 7. Comparison between Ne,S and Ne,C for the entire SQ day of  August 20, 2018 in terms of  P=Ne,S/Ne,C ratio  along dayside (panel a) and nightside semi-orbits (panel b); blue histograms represent the distribution of uncorrected P values; red histograms represent the same  distribution  after the application of our algorithms.*

## 5 Conclusions

This paper analyses the features and operation modes of Lps on board Swarm satellites, showing the occurrence of possible biases in the measurement of $N_e$ that cannot be neglected. We have identified two different effects induced on plasma surrounding the probes if a fixed negative polarization is applied, as it is the case for Swarm, for which electron density is evaluated in the ion collection region at a fixed voltage value of -2.5 V:

1. The modification in particle collection caused by plasma-sheath enlargement;
2. The possible occurrence of a strong electric field whenever satellite potential becomes significantly different from sensor potential.

For both effects, compensation algorithms able to evaluate and mitigate such biases are presented and the relevant corrected density are compared to *in-situ* observations from the Lp on board CSES-01 satellite.  August 2018 orbits have been selected for the comparison because of the close local-time conjunctions between the orbit of CSES01 and Swarm lower pair's.  Swarm electron density, corrected taking into account the above described effects, shows a remarkable agreement with the homologous measurement by CSES-01 for both nightside and dayside portions of the orbits.



The examples shown in Figure 4, where two $N_e$ time series are reported showing very good agreement after the application of correction algorithms to Swarm density, are fully representative of all orbital comparisons for August 20, 2018. In addition, a first attempt of statistical comparison between the two measurement sets is reported in Figure 7, where one can spot density ratio shifting towards unity after correction, especially for measurements taken along the dayside portion of any orbit.

Of course, more statistical data are needed in order to better characterize differences and similarities of the two sets, considering that August 20, 2018 was a quiet day in terms of geomagnetic activity, and it would be interesting to observe to what extent the effects described above can affect plasma parameters in the environment surrounding the probes during events of intense ionospheric activity.

Finally, it would be very important to build a *corpus* of consistent measurements of ionospheric parameters, especially
electron density, taken by multiple missions covering the same range of ionospheric layers: this would allow to highly improve the global coverage rate needed to feed ionospheric models, such as IRI and especially topside IRI (IRI-UP), which right now heavily draw on Swarm data as input [e.g. Bilitza et al., 2017; Pignalberi et al., 2018, and ref. therein].

**Aknowledgments**

Authors thank CSES-01 Lp PI, Dr. Liu Dapeng, and Lp developer, Dr. Guan Yi-Bing, involved in CSES-01 Langmuir probe experiment. The extended Swarm Lp dataset used in this study is available at http://earth.esa.int/swarm/. The authors also thank Dr. M. Piersanti for his suggestions for the CSES-01 LP data analysis. This work was supported by the Italian Space Agency in the framework of the "Accordo Attuativo n. 2016-16-H0 Progetto Limadou Fase E/Scienza" (CUP F12F1600011005).

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
