# Peer review of "Ionospheric Plasma Density Measurements by Swarm Langmuir Probes: Limitations and possible Corrections"

_Annales Geophysicae, 2019_

## Referee Comment (RC1) · Anonymous Referee #1 · 19 Sep 2019

Overall appreciation

This paper discusses two corrections that should be applied to the Swarm Langmuir probe measurements for ionospheric electron density. The paper presents an assessment of the quality improvement of the obtained results by comparing Swarm and CSES data. The results reported here constitute an important advance in using Langmuir probe data to build a consistent observation set of in situ ionospheric densities. The paper is well-structured. There is an excellent introduction to Langmuir probe theory, which allows the reader to appreciate the relevance of both proposed corrections. The paper also puts the work properly into context. The conclusions are clear. I have a

few suggestions for improving the presentation of the material. In particular, there are many small language and typographical issues in the manuscript. The paper will likely be suitable for publication after minor revision.

Questions and remarks

The section "Langmuir probe: general theory" is very instructive. Perhaps it would be good to somewhere state that it is written from the point of view of spherical Lps. Note that some ionospheric Lps have been cylindrical (e.g. Hoang et al. 2017, https://doi.org/10.1088/1361-6501/aa87e1; Ranvier et al. 2017, https://doi.org/10.1109/TPS.2017.2700211). It might therefore be interesting to explain how the conclusions from this section would be modified for the case of cylindrical probes.

In the Swarm-CSES comparison, the authors find that – after applying the corrections – the agreement becomes much better, but that there is a difference between the dayside and the nightside situation. A major difference between the dayside and nightside ionosphere is in the electron temperature. This therefore prompts questions about the Te that the authors have used in the correction. How has it been obtained? On lines 214-215 the authors refer to the difficulty that this correction requires a good value for Ne and Te up front. It would be useful to provide at least an estimate of the sensitivity of the correction factor to uncertainties on the given Te.

Figures 4, 6 and 7 compare the original densities to the corrected ones. There is no way to separate the respective roles of both corrections separately, while this would give valuable information about the usefulness of each correction individually. This could easily be provided, for instance, by expanding the histogram plots of figure 7.

The paper very strongly focuses on Ne as the prime result of the Lp measurements, and how it can be improved. But what about Te? Some comments would be appreciated.
Minor issues

Title: drop "possible"

17: the concept "voltage ripple" might be familiar only to a limited public; reformulating this to avoid this technical terminology could make the abstract more accessible for a broader audience

27: represents -> represent

29: mounting -> carrying

30: 470 km : I presume that this is the orbit altitude? What about the spacecraft separation between the pair?

35: in -> by

36: Lpbased -> Lp-based

37-38: The references given here focus on ionospheric density applications and compare the Lp data with remote sensing data. It could be worthwhile to also mention comparisons done in other domains between Lps and in situ plasma data (from plasma spectrometers or wave analysers), such as the one by Pedersen et al. 2008, https://doi.org/10.1029/2007JA012636.

56: mounts -> carries

66: Importantly, the conjunction is also in altitude. Perhaps it is sufficient here to just say "in conjunction with . . ."

75: drop the superfluous hyphen

85: with respect to -> with respect to the

111: thin sheath condition -> the thin sheath condition

111: thick sheath regime -> the thick sheath regime

170: Can you provide an order of magnitude error due to ignoring H+?

180: particles -> "particle's" or "particles'"

185: compensation -> Compensation

198: unmatched parenthesis? Also, this expression does not contain S_h which you explain on the next line

213: something is wrong with the construction of the sentence. Replace "results" by "is"?

228: Due to thermal velocity -> Due to the thermal velocity

228: compared to plasma potential -> compared to the plasma potential

238-255: It would clarify the explanation given here by adding a figure that sketches the configuration and a typical ion trajectory.

255: Oxigen -> Oxygen

255: Could you indicate how big the correction term is relative to Rp?

294: algorithm for sheath effect -> algorithm for the sheath effect

297: with respect to fixed probe potential -> with respect to the fixed probe potential

300: explains -> explain

300: The argument presented here focuses on the extent to which both spacecraft are at the same geographic longitudes. As the problem involves plasma density, wouldn't one rather have to make the argument for geomagnetic longitudes?

336: a strong electric field -> a strong probe-spacecraft electric field

339: are -> is

Varia:

[Figure]

Please be consistent, sometimes you use "in-situ", sometimes "in situ".

Idem for V_s/c and V_S/C.

Note that different citation styles are used throughout the text: sometimes as (. . .), sometimes as [. . .], sometimes with author first name initials and sometimes without, . . .

There is a typesetting problem with variable left text margin width.

There is also a problem with the symbol for the Debye length.

Section numbering is missing in the submitted manuscript (although such section numbers are sometimes referred to).

---

## Short Comment (SC1) · 30 Sep 2019

We compared the in situ plasma observation between CSES LAP and Swarm LP, and analyzed the possible reason why CSES LAP underestimates Ne. The following is the paper: Preliminary validation of in situ electron density measurements onboard CSES using observations from Swarm Satellites. https://doi.org/10.1016/j.asr.2019.05.025. We speculate that the possible reason may be the lower Ie current, as the estimation of Ne obtained by the other probe on CSES is larger than the one discussed in the paper. Therefore, we think calibration of CSES LAP data is a necessary work. Anyway, further study of the data is necessary.

---

## Short Comment (SC2) · 2 Oct 2019

Manuscript: angeo-2019-136 Title : Ionospheric Plasma Density Measurements by Swarm Langmuir Probes: Limitations and possible Corrections Authors : Piero Diego, Igino Coco, Igor Bertello, Maurizio Candidi, and Pietro Ubertini

=================================================================

This manuscript is definitely an interesting paper as it rightly points out some potential weaknesses of Swarm's Langmuir probe. This concerns in particular its mounting at the front bottom side, the relatively small stubs and therefore small distance ($\sim$10 cm) from the spacecraft body, which locates them most likely within the shielding layer

(Debye-sheath) of the spacecraft, at least during some parts of the orbit.

The paper contains however some serious flaws or misinterpretations despite the good and detailed introduction of the theoretical basis of the Langmuir probe measurement principle in OML approximation. Together with a quite questionable comparison to CSES-01 Langmuir probe observations at a roughly coincident orbit in about the same meridional plane, the conclusions drawn are very doubtful.

The Swarm Langmuir probe plasma density measurements are certainly much better and reliable, than this paper tries to suggest. Several studies of comparisons with radar measurements during direct overflights (e.g., Lomidze et al., 2017, also cited in the paper) or the systematic comparison with the International Reference Ionosphere (IRI) have shown, that the relative error is of the order of a 10-15% UNDERestimation. This study exemplifies with a comparison to CSES-01 measurements during August 2018 about half to one order of magnitude (factor $\sim$4 and more) OVERestimation (Fig. 4).

The key points of the study are the two items on the actual current collection of the Langmuir probes listed and discussed in section 3. They represent a trial to quantify the effects of the probe's sheath effect and the influence of the spacecraft body and its surrounding potential distribution on the measurements.

A sheath size of about 5 Debye lenghts (line 166), referred to the papers of Chen (1984, 2001), and the deduction of a corresponding Correction Factor CF within equations (5) to (7) is certainly an overestimation for the context of the Swarm satellites. The equations for the CF are hardly comprehensible, but rely probably on the papers cited. They might be valid for Langmuir probes at rest in a laboratory environment (which is the object of study in Chen's book of 1984 and the paper of 2001), but not for a supersonic plasma stream in the OML approximation, that is applied for the Swarm LP data analysis.

The second effect to be corrected, is similarly questionable. It is founded with an

electric field, that "is created between the probes and the S/C body" (lines 207-208). This electric field is understood as the difference between the fixed bias voltage in the ion current regime and the spacecraft potential, as measured by the LP. To illustrate that electric field, Fig. 3 shows their variation during an example of one daytime and one night-time orbital part.

Contrary to the author's statement, the bias potential is meant with regard to the satellite ground potential, i.e., S/C body's floating potential, fixed to -2.5 V (plus/minus the small voltage of the harmonic mode). It is quite cryptic, why the authors think, that this leads to an "unbalanced enhancement of the ion flux toward the probes" (lines 213-214). The estimations of the time of flight (ToF) of the ions between the LP and the S/C body (line 215- 216) are irrelevant, as the LPs are mounted at the bottom side, the bulk flow therefore perpendicular to this distance line. Any movement toward (or away) from the probes should consider the accelerations of the ions within the potential distribution around the spacecraft and the probes.

A plasma density of ~1E5 cmˆ-3 at ~510 km during daytime afternoon hours at low latitudes (CSES-01 in Fig. 4b) are assumed to be correct. The Swarm-A measurements at 440-460 km and approximately the same local time are corrected toward this value, although the height difference of about 50-70 km alone (disregarding the local time and latitude/longitude differences) suggest already a difference in density of up to a factor two, as it is almost one scale height (at least during nighttime).

Finally a few minor remarks.

The plots and/or the headlines of Fig. 4a,b and Fig. 5 seem to be messed somehow. Fig. 5 shows descending orbits, which was a daytime (~15 LT) passage between, e.g., between ~ 12:30 and 13:15 UT for Swarm A. The time labels in the headlines of Fig. 4a,b should be switched.

Lines 26-27: What do you mean by "free polar orbit"? Maybe: "near polar orbit"?

Line 68: The reference is Diego et al., 2017a or 2017b?

Lines 147-148: The "amount of H+ contamination" is not increasing toward high latitudes and geomagnetically active periods, rather on the contrary.

Line 251: "electronic density" -> electron density

---

## Referee Comment (RC2) · Anonymous Referee #2 · 3 Oct 2019

This paper presents analysis of potential deficiencies in the electron density measurements made by the Langmuir probes (LP) located on the Swarm satellites. After an extensive and very detailed description of the LP operation (very good and appropriate), the authors present the actual data for daytime and nighttime measurements of a single orbit. They compare the data with similar data collected on-board the Chinese satellite CSES-01, which crossed the equator at about the same local time. The comparisons show reasonable agreement between the instruments, provided that the Swarm data are corrected by two factors: 1) the plasma sheath widening effect on particle collection (more critical at high latitudes with larger Debye lengths) and 2) variations in the floating potential of the probe with respect to the satellite, along its trajectory. The data

presented indicate that the corrections have to be significant; reduction by a factor of 5. The paper is important in that a significant Swarm data base is available and the data can be used, and has already been used, for various ionospheric studies.

I have serious concerns about this paper and I cannot recommend it for publication. Although the paper is well written and has many good components, the major results reported are highly unexpected and raise a number of questions. I hope my comments below will help the authors to improve the manuscript and re-submit the paper.

1). The most critical question is: what is the final conclusion of the paper? A validation paper needs a clear conclusion and I did not find one.

2). The paper cites the validation study by Lomidze et al. (2018) but never discusses the tremendous differences in the correction coefficients in that study and in the present paper. Lomidze et al. (2018) showed that the Swarm densities agree reasonably well with ionosonde and ISR measurements at middle latitudes, by considering two classical instruments that have never been questioned in terms of the data quality. The authors should explain their reasoning as to why the paper by Lomidze et al. (2018) can be ignored and new correction factors for Swarm electron density data need to be introduced.

3). Validation of the Chinese data is an open and unresolved issue. For this kind of paper, co-authorship, contributions or at least confirmation from the CSES-01 team are absolutely required, especially in view that serious doubts are expressed with respect to the Swarm data quality on the ESA website.

4). The paper clearly mentions that one measurement might be an anomaly and statistical assessment is needed (Lines 41-44). Then the value of the analysis of a single event is questionable. Statistical analysis is required to make a confident judgment.

5). Since the proposed corrections are quite drastic, does that mean that individual researchers cannot use Swarm electron density data directly from the ESA website?
This needs to be commented on in the paper.

6). Since the proposed corrections are straightforward and follow the equations given, why not evaluate new "corrected" densities versus those on the ESA website? Comparisons with various models is another possible and reasonable step. These actions would be instructive in assessing by how much the Swarm data would have to be rescaled, on average.

7). The Debye length factor is more important at high latitudes (Line 167). I expected that the authors would seek Swarm-CSES-01 conjunctions at high latitudes and not at equatorial latitudes. It is understandable that this might not be easy, or may be impossible. Comments are required on this issue as well.

8) In Figure 4, data for the nightside and the dayside are presented. I expected that the "point-by-point" comparison would be done on Figure 6, not only for the dayside but for the nightside as well. The reasons for the difference in agreement need to be discussed.

Technical issues:

I found that the authors use excessively commas, it makes reading more confusing. Below I report several items that the authors might wish to fix.

L15: extent

L23: LPs need to be spelled out as this is first appearance

L23: represent

L26,27: remove "free"

L36,37: remove "even though"

L38: ISR and ionosondes (these are different instruments)

L36: all altitudes

L39: all latitudes

Eq (1): "k" needs to be introduced here and dropped in a couple of times later. For eq.(1), reference is required in line 80.

L90-95. Split on several sentences

L108: deviation

L109: et al.

L115: Reference is required here

L140: Knudsen et al showed how . . .

L144: q and V have been already introduced

L149-154: These statements do not prove that LP works correctly

L169: meaning of the statement is not clear. "saturation" comes out of blue

L179: fix SQRT

L180: "K" has already been introduced

L230: efficiency »> validity

L255: 1984;

L299: remarkable »> reasonable

---

## Author Comment (AC1) · 10 Oct 2019

Dear Dr. Xiuying Wang, Thank you for the comment. The paper you have suggested ascribes the Ne uncertainty to that relevant to the detection of plasma potential in the classic Langmuir procedure. This is a well-known issue in plasma diagnostic. In addition, the authors quoted the work of Godyak and Alexandrovich (Comparative analysis of plasma probe diagnostic, 2015) that estimates the Ne uncertainty up to one order of magnitude in some extreme conditions. This issue is addressed by Rui Yan et al., (The Langmuir probe onboard CSES: data inversion analysis method and first results, EEP, 479-488, 2018, doi:10.26464/epp2018046), who introduce the iterative method

to minimize the plasma potential computational error that reduces the Ne uncertainty as well. Moreover, the use of a large probe (i.e. collecting larger current from plasma) allows a better identification of the inflection point in the characteristic curve. In our work, we used data collected by the larger CSES Lp probe with 5 cm diameter rather than the one with 1 cm. The very good agreement between the time series trend of Ne and Te of Swarm and CSES is, in our opinion, a proof that both instruments work properly but there is some lack in the accuracy of the current collected. In fact, as known, such issue would affect the Ne (that depends on the current level) rather than Te (that depends on the shape of the I-V curve) that shows indeed a high correlation. For these reasons our paper focuses on identifying the reasons for the large discrepancies (a factor from 4 up to 20) based on physical effects on the particles collection by the Swarm probe. We consider such physical effects of greater importance in determining the discrepancy between measurements. Best regards. Piero Diego

---

## Author Comment (AC2) · 10 Oct 2019

Dear Dr. Forster, Thanks for the detailed comments that may help us to substantially improve our analysis. The aim of our work is not to bend Swarm Ne to that of CSES, but is to find the reasons why the two instruments show such a high discrepancy in the absolute values while they track each other in the shape of the time series along the orbit (reaching a very good agreement when at almost same LT). Of course, the expected difference due to the different altitude (about 35% from IRI values) and orbits have been taken into account, as reported in Sect. 4 (Fig.5 and relevant comment). The results obtained by Lomidze et al. are not a matter of discussion for us, because, even

if they show consistent agreement between the measurements and the models used, we think that in-situ observations should be treated separately since these are obtained with similar procedures, and these are quite different from those used for the calibration made by Lomidze et al. The agreement in Lp measurements could be a simpler matter once the actual value of collected current is determined. For this reason, we consider very important to find a way to match in-situ measurements in order to provide the proper reference values for ionospheric plasma models. Of course, also CSES Lp data may need a calibration review, especially for what concern the plasma potential detection (e.g. Rui Yan et al., The Langmuir probe onboard CSES: data inversion analysis method and first results, EEP, 479-488, 2018, doi:10.26464/epp2018046). However, it is our feeling that the harmonic mode of Swarm Lp, and in particular its negative bias, could produce more important interferences in the ion collection described in OML theory. With reference to the shape and effect of a sheath around the probe, we appreciate your suggestion and the one of Dr. Buchert (private communication) which warn to consider the Chen reference and the sheath presence itself in case of fast moving objects in the plasma (i.e. S/C velocity). Still we believe in the sheath presence and in its effect on ion collection but we would rather suggest the specific results of Whipple (Potential of surface in space, 1981) for a revised evaluation of the actual current collection. To examine in depth your warnings, we have chosen Whipple (1981) results that summarize the condition in which the sheath effect on ion collection is applicable. In fact, if the relative velocity between S/C and plasma is much greater wrt the thermal velocity, this effect becomes very small with respect to the static scenario we suggested. The increase in current collection, therefore, should be within a few tens of % more than that collected by the probe disk area. The magnitude of this current enhancement is also in agreement with that suggested by Buchert. About the "electric field effect" induced by the S/C-Lp potential difference, we thank you for the information about the reference GND that we misinterpreted. If we understand well, such potential difference is fixed at -2.5V along the orbit. In our paper, the word "unbalanced" means that the cross-section grows only toward the S/C but not in the outer direction. We

estimate the ToF to identify how long the electric field acts pushing down towards the probe the ions that are travelling in the space between S/C and Lp. This path can be roughly considered to start at the edge of the S/C sheath and to finish while crossing the stub of the Lp.

Anyway, as the sheath is the charge layer that shields electric fields around a polarized body, the electric field is confined inside the sheath. This means that, as long as the sheath is small, the probe polarization effects are localized very close to the probe and, in addition, they are already described by OML theory. In such cases corrections are not applicable. Nevertheless, when the plasma density becomes lower (e.g. at higher latitudes or inside plasma bubbles or Travelling Ionospheric Disturbances,...) and the Debye length consequently increases, the probe and the S/C sheaths could melt and the electric field between them is no longer shielded, giving rise to saturation effects due to ions amount inside the enlarged global sheath. Unfortunately, this scenario appears to be very hard to describe with a simplified model and so the relevant current collection enhancement. All those issue imply that the paper cannot be corrected in its current version but it needs a complete revision. We aim to carry such study out in cooperation with the Swarm Lp developer team in order to improve the feedback rate and the quality of the analysis. We decided therefore to withdraw the paper in its current form. Best regards. Piero Diego

---

## Author Comment (AC3) · 10 Oct 2019

Authors want to thank the Referee for the general comment and they appreciate the suggestions that surely will be applied in a new realise of the paper. The analysis of cylindrical Lp, even if interesting for a general description of Lp, it appears a little bit out of theme for our paper since both satellites have spherical one. A short description of Te used for the calculation of the Debye length along the orbit will be included in future revisions. Anyway, since the two time-series of Te are in very good agreement, no differences have been found in the correction factor when applying CSES or Swarm values. We could state that discrepancies found in Ne, rather than in Te, suggests that

the two instruments work properly but at different collected current level. Best regards
Piero Diego

Interactive
comment

---

## Author Comment (AC4) · 10 Oct 2019

We want to thank the Referee for the general comment and the suggestions. The aim of our work is to find the reasons why Swarm and CSES plasma densities show a high discrepancy in the absolute values while they track each other in the shape of the time series along the orbit (reaching a very good agreement when at almost same LT). In particular, we are interested in the variability of such discrepancy that appear to be related to the Debye length variation along the orbit and during plasma depletion occurrences. Our thesis is that both instruments produce reliable data but could need additional calibrations as well. Even if systematic differences could be

ascribed to various effect (i.e. data inversion algorithms, altitude difference, ...) we aim that some anomalous increases in the Swarm/CSES Ne ratio should be investigated. Anyway, we cannot state that the entire data set of Swarm is not valid and has to be corrected. Results obtained by Lomidze et al. show consistent agreement between the measurements and the models used, but we think that in-situ observations should be treated separately since these are obtained with similar procedures, and these are quite different from those used for the calibration made by Lomidze et al. Thanks to suggestions from Swarm team, we have deepened some aspects of our analysis finding that the sheath effect is very small in case of fast moving object (only about 50% more than that of probe cross section) and also the electric field induced by S/C presence is usually confined inside the probe sheath (thus it is already described by OML theory). On the other hand, the case in which the sheath radius is comparable with the probe stub (and the possible melting between probe sheath and S/C sheath) need to be furtherly investigated. We think the paper cannot be upgraded starting from its current version but it needs a complete revision, thus we decided to withdraw it. Best regards.

Piero Diego

---

## Short Comment (SC3) · 11 Oct 2019

There is another reason why we conclude that Ne is underestimated by LAP on on-board CSES. That is Ne estimated from another probe, the small one, is larger than the formal published Ne data, which I mentioned in the initial comment, but you neglect it. The most possible reason as pointed out is underestimation of the Ie. Ie from both probes is lower than it should be, but Ie from the larger one loses more current, therefore a lower Ne estimation.

---

## Author Comment (AC5) · 5 Nov 2019

Authors want to thank for the detailed comments that will help us to substantially improve our analysis. The aim of our work is to find the reasons why the Langmuir probes on board CSES and Swarm satellites show such a high discrepancy in the plasma density measurements while they track each other in the shape of the time series along the orbit (reaching a very good agreement when at almost same LT). Of course, the expected difference due to the different altitude and orbits should be considered. Although Swarm calibration obtained by Lomidze et al. shows consistent agreement between the measurements, we think that in-situ observations should be

treated separately since these are obtained with similar procedures. The agreement in Lp measurements could be a simpler matter once the actual value of collected current is determined. For this reason, we consider very important to find a way to match in-situ measurements in order to provide the proper reference values for ionospheric plasma models. Of course, also CSES Lp data need a calibration review, especially for what concern the plasma potential detection (e.g. Rui Yan et al., The Langmuir probe on board CSES: data inversion analysis method and first results, EEP, 479-488, 2018, doi:10.26464/epp2018046). However, it is our feeling that the harmonic mode of Swarm Lp, and in particular its negative bias, could produce more important interferences in the ion collection described in OML theory. With reference to the shape and effect of a sheath around the probe, we appreciate suggestions from Swarm Lp developer which warn to consider the Chen reference and the sheath presence itself in case of fast moving objects in the plasma (i.e. S/C velocity). Still we believe in the sheath presence and in its effect on ion collection but we would rather suggest the specific results of Whipple (Potential of surface in space, 1981) for a revised evaluation of the actual current collection. To examine in depth such warnings, we have chosen Whipple (1981) results that summarize the condition in which the sheath effect on ion collection is applicable. In fact, if the relative velocity between S/C and plasma is much greater wrt the thermal velocity, this effect becomes very small with respect to the static scenario we firstly suggested. The increase in current collection, therefore, should be within few tens of % more than that collected by the probe disk area. We also discussed the "electric field effect" induced by the S/C-Lp potential difference, that is fixed at -2.5V along the orbit, to quantify an "unbalanced" probe cross-section (meaning a cross-section growing only toward the S/C but not in the outer direction). We estimate the ToF to identify how long the electric field acts pushing down towards the probe the ions that are travelling in the space between S/C and Lp. This path can be roughly considered to start at the edge of the S/C sheath and to finish while crossing the stub of the Lp. Anyway, as the sheath is the charge layer that shields electric fields around a polarized body, the electric field is confined inside the sheath. This means that, as

long as the sheath is small, the probe polarization effects are localized very close to the probe and, in addition, they are already described by OML theory. In such cases corrections are not applicable. Nevertheless, when the plasma density becomes very low (e.g. at higher latitudes or inside plasma bubbles or Travelling Ionospheric Disturbances) and the Debye length consequently increases, the probe and the S/C sheaths could melt and the electric field between them is no longer shielded, giving rise to saturation effects due to ions amount inside the enlarged global sheath. Unfortunately, this scenario appears to be very hard to describe with a simplified model and so the relevant current collection enhancement. We are currently performing additional analysis aiming to explain and quantify the two different kind of discrepancy level observed between Swarm and CSES that are, the average difference, and the extreme difference that occur at very low density (e.g. below 109 m-3). The average ratio (about 4) could be addressed to; i) different altitude (about 35%), ii) sheath increasing collection of Swarm (within 50%), and iii) CSES plasma potential computation uncertainty (about 50%). On the other hand, the extreme discrepancy evaluation needs to model the electric field topology inside the melted sheath, still depending to density level as shown during exceptional plasma depletions where the discrepancy level reached about 3 order of magnitude. All those issue imply that the paper cannot be corrected in its current version but it needs a complete revision. We aim to carry such study out in cooperation with the Swarm Lp developer team in order to improve the feedback rate and the quality of the analysis. We decided therefore to withdraw the paper in its current form.